# Functional Space Variational Inference for Uncertainty Estimation in Computer Aided Diagnosis

**Pranav Poduval**                                          PRANAV97.PODUVAL@GMAIL.COM
**Hrushikesh Loya**                                         LOYAHRUSHIKESH@GMAIL.COM
**Amit Sethi**                                              AMITSETHI@GMAIL.COM
*Department of Electrical Engineering, Indian Institute of Technology, Bombay*

## Abstract

Deep neural networks have revolutionized medical image analysis and disease diagnosis. Despite their impressive performance, it is difficult to generate well-calibrated probabilistic outputs for such networks, which makes them uninterpretable black boxes. Bayesian neural networks provide a principled approach for modelling uncertainty and increasing patient safety, but they have a large computational overhead and provide limited improvement in calibration. In this work, by taking skin lesion classification as an example task, we show that by shifting Bayesian inference to the functional space we can craft meaningful priors that give better calibrated uncertainty estimates at a much lower computational cost.

**Keywords:** Bayesian approximation, uncertainty estimates, calibration, skin lesions

## 1. Introduction

In computer-aided diagnosis, AI models must not only be accurate, but they should also indicate when they are likely to be incorrect. For instance, control should be passed on to human doctors when the confidence of a neural network for disease diagnosis is low. Model calibration is the degree to which a models predicted probabilities reflect the true correctness likelihood. Calibrated confidence estimates are also important for model interpretability and they provide a valuable extra bit of information to establish trustworthiness with the user. This is important for deep neural networks, whose classification decisions are often and difficult to interpret.

It is well known that popular neural network frameworks only provide a point estimate of the true underlying distribution. Furthermore, the typical classification setting of training the softmax output layer using cross-entropy loss typically gives over-confident (low entropy) class probability mass distributions, even when there is a classification error. This is especially concerning for training on medical datasets that are often relatively smaller and suffer from severe class imbalance (Esteva et al., 2017). In other words, the popular deep learning models give poorly calibrated uncertainty estimates for cases that are ambiguous, or difficult, or *out-of-distribution* (OOD), including those from a new class.

Bayesian modelling offers a set of tools to reason about uncertainty. Existing Bayesian approaches involve approximate inference using either Markov Chain Monte Carlo (Neal et al., 2011) or variational inference methods, such as dropout (Gal and Ghahramani, 2016). This idea has attracted attention of the medical community to ensure that difficult cases for computer-aided diagnosis are duly flagged for review (Laves et al., 2019). Since most

Bayesian neural networks (BNNs) have their prior defined on the weight space, the regularization caused by these prior is not able to calibrate the network output, nor do these priors explicitly make the model under-confident on the OOD samples. We show that by performing variational inference on the functional space we can craft a prior that is able to simultaneously calibrate the network as well as ensure the the recognition of OOD samples as more uncertain. Our method is also significantly less computationally expensive as compared to Bayesian or frequentist approaches. Although our method shares some similarities with Evidential Deep Learning (EDL) (Sensoy et al., 2018), it has been derived from a variational Bayesian framework and it can distinguish distributional versus data uncertainties (shift in distribution versus class confusion, respectively), unlike EDL.

## 2. Proposed Method

For classification among $K$ classes, deep neural networks represent a function $f_\theta : X \to \mathbf{p} \in [0,1]^K$, where $X$ represents the input, and $\mathbf{p}$ represents a probability mass function such that $\sum_{i=1}^K p_i = 1$. The output distribution $p(Y|X, \theta) = \text{Cat}(Y|\mathbf{p})$. A prior on $\theta$ implicitly defines a prior measure on the space of $f(X)$, denoted as $p(f)$. Priors of convenience on $\theta$, such as a fully factorized Gaussian, are often used, and it is difficult to formulate a prior on the weight space that is informative in the sense that it leads to high uncertainty on OOD examples. We therefore define a uniform prior on the $K$-dimensional unit simplex for the functional space, such that $p(f) = D(\mathbf{p}|\langle 1, \ldots, 1\rangle)$ (completely uncertain prior). While it seems intuitively satisfying to have a model that is not biased towards "over-confident" outputs (towards which the usual cross-entropy loss is severely biased), we also empirically show that such a uniform prior gives well-calibrated outputs.

Given the training data $D = (X^D, y^D)$ and the test points $(x^*, y^*)$ we have:

$$p(y^*|x^*, D) = \int p(y^*|\mathbf{p}) \, p(\mathbf{p}|x^*, D) \, d\mathbf{p} \tag{1}$$

We assume $p(y^*|\mathbf{p}) = \text{Cat}(y^*|\mathbf{p})$. We further assume that the neural network estimates a Dirichlet distribution $\text{Dir}(\mathbf{p}|\alpha)$ with $\alpha > 0$, as done by Sensoy et al. (2018), because of its analytical tractability. In other words, unlike for a standard neural network where $\mathbf{p} = f_\theta(x)$ is the point estimate output, in our case $\text{Dir}(\mathbf{p}|\alpha) = q_\theta(f(x))$ is the marginal functional distribution. This is similar to how a Gaussian process has a multivariate Gaussian as its marginal distribution.

The true functional posterior $p(f|D)$ is intractable, but it can be approximated by minimizing the functional evidence lower bound (fELBO) as done by (Sun et al., 2019):

$$\mathcal{L}(q) = -\mathbb{E}_{q(f)}[\log p(y^D|f(X^D))] + \text{KL}[q(f)||p(f)] \tag{2}$$

The second term in Equation 2 is the functional KL divergence, which is hard to estimate. Therefore, we shift to a more familiar metric, the KL divergence between the marginal distributions of function values at finite sets of points $\mathbf{x}_{1:n}$. (Sun et al., 2019) has shown:

$$\text{KL}(q(f)||p(f)) = \sup_{\mathbf{x}_{1:n}} \text{KL}\left[q(f(\mathbf{x}_{1:n})||p(f(\mathbf{x}_{1:n})\right] = \sum_{i=1}^n \text{KL}[\text{Dir}(\mathbf{p}|\alpha_i)||\text{Dir}(\mathbf{p}|\langle 1, \ldots 1\rangle)] \tag{3}$$

A more relaxed way of sampling these measure points" $\mathbf{x}_{1:n}$, is to assume $\mathbf{x}_{1:k} \sim X^D$ (training distribution) and $\mathbf{x}_{k+1:n} \sim c$ where $c$ is a distribution having the same support as the training distribution, which could be OOD samples, that can be forced to be more uncertain. This approach is similar to (Hafner et al., 2018), (Malinin and Gales, 2018).

We get a closed form solution for the first part in Equation 2 by assuming $y$ to be a one-hot vector as follows:

$$\mathcal{L}_1 = \int \left[ \sum_{i=1}^{K} -y_i \log p_i \right] \frac{1}{B(\alpha)} \prod_{i=1}^{K} p_i^{\alpha_i - 1} d\mathbf{p} = \sum_{i=1}^{K} y_i \left( F(\sum_{j=1}^{K} \alpha_j) - F(\alpha_i) \right) \qquad (4)$$

$F(.)$ is the digamma function. To measure calibration of the proposed model we group predictions $p \in [0, 1]$ into $M$ bins each of size $\frac{1}{M}$, and let $B_m$ be the set of indices of samples whose prediction confidence falls into the interval $(\frac{m-1}{M}, \frac{m}{M}]$ for $m \in \{1, \ldots, M\}$. Now we define accuracy of $B_m$ as acc$(B_m) = \frac{1}{|B_m|} \sum_{i \in B_M} 1_{\{\hat{y}_i = y_i\}}$, where $\hat{y}$ is the predicted outcome with confidence $p$, and $y$ is the true label. Similarly, we define the average confidence as conf$(B_m) = \frac{1}{|B_m|} \sum_{i \in B_m} p_i$. For perfect calibration we expect acc$(B_m) = $ conf$(B_m)$. In order to quantify how well calibrated our networks are, we use Expected Calibration Error (ECE) $= \sum_{i=1}^{M} \frac{|B_m|}{n} |$acc$(B_m) - $conf$(B_m)|$ as the metric. Note ECE $= 0$ for perfect calibration.

## 3. Results

We applied our method to the problem of skin lesion classification, using the HAM10000 dataset (Tschandl et al., 2018). ResNet 50 architecture optimized by Adam was used.

From Table 1 we can see that although standard Bayesian approaches do help calibrate the model, our method has a significantly lower ECE. That too at a much lower computational cost, approximately 25x less computationally expensive than Dropouts (monte carlo approximation) and 5x more memory efficient than Ensembles (ensemble size).

Table 1: Comparison of classification accuracy and ECE on HAM10000 dataset

| Method | Standard NN | Dropout | Deep Ensemble | Functional Space VI |
|---|---|---|---|---|
| Test Accuracy | 84.38% | **86.32%** | 85.21% | 84.84% |
| ECE (M = 15) | 7.73% | 6.39% | 3.12% | **1.17%** |

The entropy $\mathcal{H}[p(y|x, D)]$ is a measure of total uncertainty, whereas differential entropy $\mathcal{H}[D(\mathbf{p}|\alpha)]$ is a measure of the distributional uncertainty.(See Appendix A for more details)

## 4. Conclusions

We proposed a novel Bayesian NN framework whose prior explicitly forces OOD samples to become unconfident as well as allow us to estimate uncertainty analytically at test time, without needing approximate or expensive algorithms. We have also shown that our model gives well-calibrated uncertainty outputs, which can increase patient safety and assist a transfer of AI systems into clinical settings by including trustworthiness as a design factor in machine learning models for medical diagnosis. Our method is also significantly more computationally efficient making it a more viable option for resource-constrained problems.

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

## Appendix A. Quantifying Uncertainty

We use two measures to estimate uncertainty – differential entropy and output entropy. The output entropy is a measure of the total uncertainty, where as the differential entropy is a good measure of distributional uncertainty. Output entropy is high whenever we encounter overlap between classes or we encounter samples from OOD. On the other hand, the differential entropy is high only when we encounter OOD samples and remains low even in case of data uncertainty (Malinin and Gales, 2018).

Output entropy is defined as:

$$\mathcal{H}[p(y|x, D)] = -\sum_{i=1}^{K} p(y_i|x, D) \log(p(y_i|x, D)) \tag{5}$$

where

$$p(y_i|x, D) = \int p(y_i|\mathbf{p})\text{Dir}(\mathbf{p}|\alpha)d\mathbf{p} = \frac{\alpha_i}{\sum_{j=1}^{K} \alpha_j} \tag{6}$$

Differential entropy is maximized when all categorical distributions are equiprobable. i.e. when posterior $q_\theta(f(x)) = D(\mathbf{p}|\langle 1, \ldots, 1 \rangle)$, and it is defined as:

$$\mathcal{H}[D(\mathbf{p}|\alpha)] = \log B(\alpha) + (\sum_{i=1}^{K} \alpha_i - K)F(\sum_{i=1}^{K} \alpha_i) - \sum_{i=1}^{K}(\alpha_i - 1)F(\alpha_i) \tag{7}$$

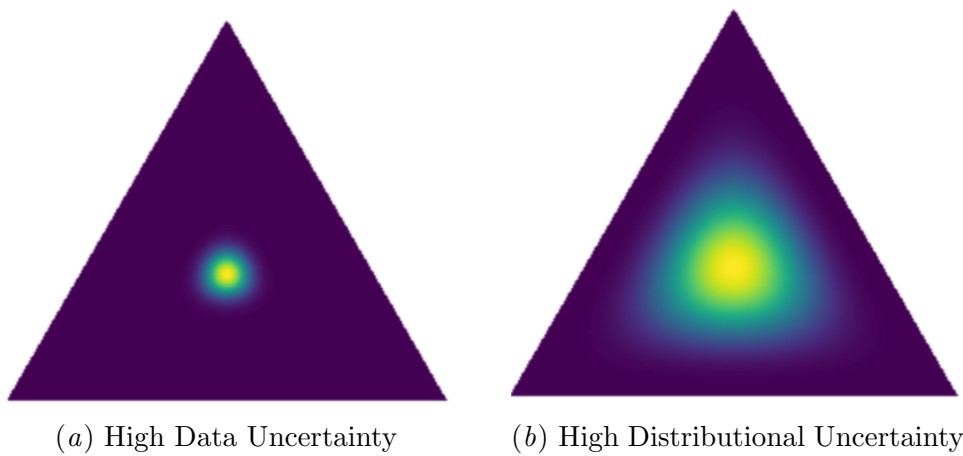

$(a)$ High Data Uncertainty          $(b)$ High Distributional Uncertainty

Figure 1: (a) implies high data uncertainty so we will have low differential entropy (b) has high distributional uncertainty so both uncertainty metrics will be high

From Figure 1 it becomes clear that our method allows us to easily distinguish between Data and Distributional Uncertainty.

## Appendix B. Additional Experiment

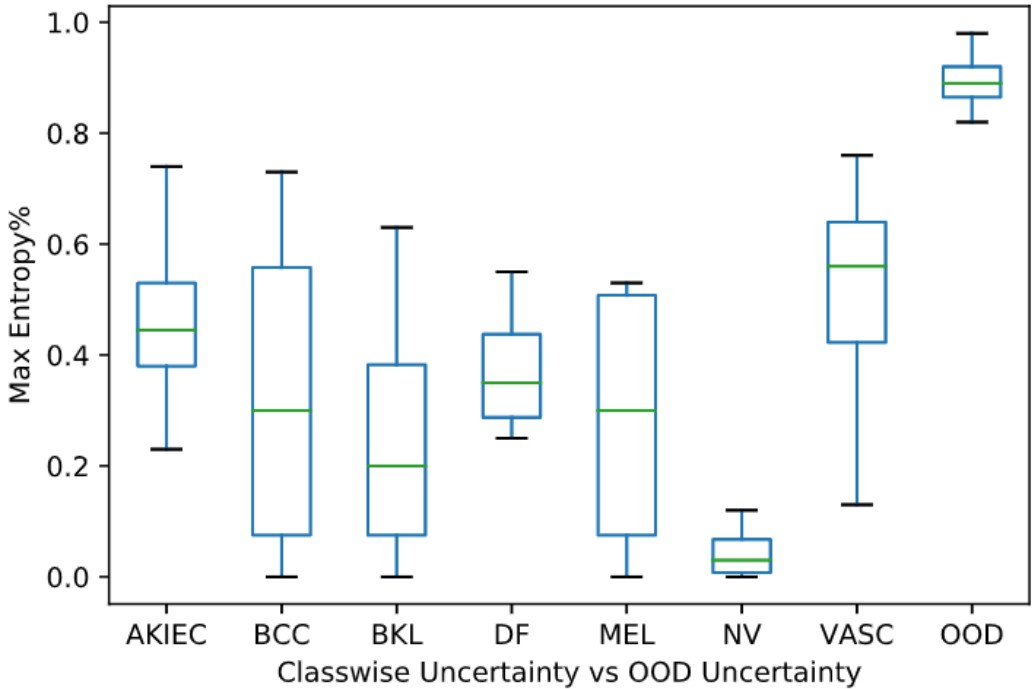

We observe our model is very confident on Nevi (NV) class, which is expected since it make majority of the dataset, this reinforces the importance of well balanced data for learning. We can also see our OOD samples can be distinctly separated from the in-class samples. The OOD sample used for training and testing are from different distributions. For simplicity we used Gaussian Distribution for training OOD samples and Uniform Distribution for testing OOD samples. Ideally more complex techniques should be used for generating OOD samples on the decision boundary (Hafner et al., 2018).

