# OpenReview forum: "Functional Space Variational Inference for Uncertainty Estimation in Computer Aided Diagnosis"
_MIDL.io/2020/Conference — MIDL 2020_

### Official Review · AnonReviewer3 · 2020-02-26
**Inference in the function space for well-calibrated DNNs. Claims are not supported by experiments and results.**

**Rating:** 2
**Confidence:** 4

**Review:**

The paper studies the predictive uncertainty estimation for medical diagnosis. Basically, they move from the weight space view to the function space view and run their proposed inference method directly on functions. The main idea is that the DNNs can be better calibrated due to the direct modulation of functional outputs. They also claim that the OoD example will be better detected this way. While the paper is in principle interesting, I have the following concerns.

Experiments are limited and results do not substantiate the claims. No results reported for OoD detection. So, how could I just judge the performance in this regard? They also say that they can better distinguish the distributional uncertainty from data uncertainty, in comparison to Sensoy et al. (2018). But, no results... Only Table 1 reports on classification accuracy on skin lesion classification and the calibration error. However, even this comparison lacks proper evaluation. The drop in ECE seems significant but there is mention of any statistical test, despite the bold statement of 'significantly lower ECE'. Also the computational efficiency comparison between MCDO and DNN ensemble is a bit careless. How many samples were drawn for MCDO? What is the ensemble size? How can one interpret 25x or 5x gain under this scenario?

I have a few reservation in regards to writing, too. References float freely in text. Also, additional references are required since some of those sentences are derived/learned from earlier work. For example,

"Furthermore, the typical classification setting of training the softmax output layer using cross-entropy loss typically gives “over-confident” (low entropy) class probability mass distributions, even when there is a classification error [HERE]." This is especially concerning for training on medical datasets that are often relatively smaller and suffer from severe class imbalance -(- Esteva et al. (2017) -)-.
In other words, the popular deep learning models give poorly calibrated uncertainty estimates for cases that are ambiguous, or difficult, or out-of-distribution (OOD), including those from a new class [also HERE]."

ECE should also be cited since it is not invented in the current work.

In Eq.4, what is that fancy 'F'?

In summary, the paper has a good motivation and seems to go in the right direction. However, the paper is in its infancy and not very convincing in its present form.

---

### Official Review · AnonReviewer4 · 2020-03-11
**Good paper**

**Rating:** 3
**Confidence:** 4

**Review:**

The key idea in the paper is to use functional prior that is completely uncertain about prediction of any class. To achieve this , the idea of introducing Dirichlet distribution after neural network is used from Evidential Deep Learning (EDL) paper.
From table 1, it is clear that ECE is much lower for the proposed method. However, I have following concerns:
1. It is not clear why calibration is reported and not simple measures of uncertainty like variance or entropy? Also, I would be convinced that the variance would increase for out of distribution test samples because you used a prior that enforced uncertainty of all labels. Now, it is difficult to connect  use of prior and improvement in ECE.

2. What is the experimental setup? Did you train on some other dataset and test on skin lesion dataset?

3. Last line of section 1: "it can distinguish distributional versus data uncertainties". How?

Overall, the idea is fine.

---

### Official Review · AnonReviewer1 · 2020-03-13
**Interesting abstract and a suitable contribution to MIDL.**

**Rating:** 4
**Confidence:** 5

**Review:**

The work experiments with a flat prior in functional space for multilabel classification, in a Deep Learning setting. The network f_theta predicts, given the input x, the concentration parameters alpha of a Dirichlet prior on the vector of class probabilities. After marginalizing over class probabilities this yields the standard softmax on label probabilities (up to reparametrization $\alpha$ <-> $\exp{\eta}$).
The functional prior is built from the Dirichlet distribution with $\alpha=1$; and evaluated on a measurement set in the data space that suitably accounts for in- and out-of-distribution points.

I think this abstract is suitable to be presented at MIDL.

The approach is not necessarily very novel in practice, but the functional space perspective is still uncommon and interesting (including the soft-constraints induced by a prior that looks uninformative at first glance). Also, the functional viewpoint as a way to incorporate Bayesian priors in neural networks is a promising direction.

There are a few typos that can be corrected. The choice of validation is suitable and reasonably executed given the format. The entropy/ies are mentioned at the very end but not reported?

There are a few claims that do not necessarily serve the argument:
"uncertainty outputs, which can increase patient safety [...]" -> maybe not necessary to go there unless you have results?
"Our method is also significantly less computationally expensive as compared to Bayesian or frequentist approaches" -> At most it is orthogonal to being Bayesian or frequentist. The work is quite clearly using the Bayesian toolbox, including evidence lower bound computations (as per the title(!), it is an instance of variational inference).

---

### Official Review · AnonReviewer2 · 2020-03-13
**A very dense paper with potential merit**

**Rating:** 3
**Confidence:** 2

**Review:**

The paper proposes to place a Bayesian prior on the distribution generated by a deep network instead of the traditional approach in Bayesian deep learning, where the prior is placed on the weights.

The choice of short paper makes the presentation extremely dense, and it is thus very hard to thoroughly evaluate the paper. This goes for both the mathematical developments, and for the correctness of statements such as "the regularization caused by these prior is not able to calibrate the network output, nor do these priors explicitly make the model under-confident on the OOD samples.".

However, setting the readability aside, I believe the idea pursued in the paper - to define the prior on the output distribution - has merits. The authors develop the variational framework for training the network, define an evaluation criteria (the ECE measure), and validate the model experimentally on skin lesion classification.

Overall, though the paper would certainly benefit from more pages to elaborate on all aspects of the model, and though I cannot fully validate its correctness, I find the paper has potential merit and would be an interesting read for the MIDL audience.

---

### Meta-Review · Area_Chair1 · 2020-03-26
**MetaReview of Paper310 by AreaChair1**

**Rating:** 3

**Metareview:**

All reviewers agree that, despite some presentation flaws, the work pursues an interest direction and provides a new interesting perspective.  While I understand some of the more serious concerns raised by Rev. 3 in terms of lack of detail or the paper being half baked, I also think that one also needs to factor in the short paper format of the submission.

**Paper Type:**

methodological development

---

### Decision · Program_Chairs · 2020-04-11

Accept